# Gold Inks for Inkjet Printing on Photo Paper: Complementary Characterisation

**DOI:** 10.3390/nano11030599

**Published:** 2021-02-28

**Authors:** Hanuma Reddy Tiyyagura, Peter Majerič, Matej Bračič, Ivan Anžel, Rebeka Rudolf

**Affiliations:** 1Faculty of Mechanical Engineering, University of Maribor, Smetanova ulica 17, 2000 Maribor, Slovenia; hanuma.tiyyagura@um.si (H.R.T.); peter.majeric@um.si (P.M.); matej.bracic@um.si (M.B.); ivan.anzel@um.si (I.A.); 2Zlatarna Celje d.o.o., Kersnikova ulica 19, 3000 Celje, Slovenia

**Keywords:** gold nanoparticles, ultrasonic spray pyrolysis, gold inks, characterisation, inkjet printing

## Abstract

Nowadays, cost-effective, available, and flexible paper-based electronics play an essential role in the electronics industry. Herein, we present gold nanoparticles (AuNPs) as a potential raw material for gold inks in the future for such purposes. AuNPs in this research were synthesised using the ultrasonic spray pyrolysis (USP) technique from two precursors: gold (III) chloride tetrahydrate and gold (III) acetate. Synthesised AuNPs were collected in a suspension composed of deionised (D.I.) water and the stabiliser polyvinylpyrrolidone (PVP). AuNPs’ suspensions were subjected to the rotavapor process to obtain gold inks with higher Au concentration (>300 ppm). ICP-MS measurements, the size and shape of AuNPs, ζ-potential, Ultraviolet-visible (UV-Vis) spectrophotometry measurements, and scanning electron microscop y (SEM) of gold inks were carried out in order to find the optimal printing parameters. In the final stage, the optical contact angle measurements were performed using a set of polar to non-polar liquids, allowing for the determination of the surface free energy of gold inks. Inkjet printing of gold inks as defined stripes on photo paper were tested, based on the characterisation results.

## 1. Introduction

In recent years, printed paper electronics have gained attention in various applications, like electrochemical sensors, energy storage devices, solar cells, and radiofrequency identification tags [1,2,3,4,5,6,7,8,9]. Paper-based inkjet-printed flexible electronic circuits with cost-effective recyclability, and ecological features, are an alternative to the ongoing current electronic devices [10,11,12]. Different types of printing techniques are available in the market to fabricate paper electrodes, such as photolithography [13,14], screen printing [15,16], the drop-on-demand inkjet printing technique [17,18,19,20], etc. Among these, inkjet printing is one of the easy and non-contact direct ink writing methods for printing paper electrodes with high resolution and precision without a mask pattern compared to other printing techniques. For inkjet printing, conductive inks are used, composed mostly on the basis of metallic nanoparticles like silver, copper, and gold [21,22]. Especially, gold nanoparticles (AuNPs) are applied widely in electronics, sensors, probes, catalysis, and photochemical therapy, due to their unique properties [23,24,25,26]. AuNPs are, namely, highly conductive and biocompatible, besides possessing surface plasmon resonance and special optical and electronic properties, depending on their sizes and shapes [27,28]. 

Based on the presented “state-of-the-art” for printed paper electronics, our main research focus was to prepare a low cost, stable, and useful gold ink, which can easily be used for inkjet printing on different types of paper; photo paper was used for testing in this study. ultrasonic spray pyrolysis (USP) bottom-up and the rotavapor process were applied for preparation of two types of gold inks, which differed in the chemical composition of the precursor. The flexibility of the USP method allows it to produce AuNPs and, subsequently, gold ink, from various precursor salts and in differing shapes and sizes. Feasibility studies were performed for preparing and using gold ink from the USP method [27,28]. However, there were some gaps in the literature and research regarding the properties of the prepared inks and their usability in inkjet printing. The physical properties of the prepared gold inks were explained with the different characterisation techniques; in addition, their surface behaviour was explained with the help of optical contact angle and surface energy measurements in order to perform optimal printing. Finally, the quality of the printed strips was determined and characterised. This study highlights the properties of the different gold inks produced by USP, for determining the most applicable ink for printing.

## 2. Materials and Methods

### 2.1. Materials

Starting materials for the preparation of Au precursors: gold (III) chloride tetrahydrate—AuCl (trace metals basis 99.9%, Acros Organics, Nidderau, Germany) and Au acetate salt—AuAc (gold (III) acetate, Alfa Aesar, Haverhill, MA, USA). Collection medium: deionised water (D.I.) and polyvinylpyrrolidone (PVP40 Sigma Aldrich, Darmstadt, Germany). Other chemicals: sodium hydroxide (NaOH, Fisher Chemicals, Geel, Germany) and hydrochloric acid (HCl, 37%, Sigma Aldrich, Taufkirchen, Germany).

### 2.2. Methods

#### 2.2.1. Preparation of AuNPs

AuNPs were produced with a USP device located in Zlatarna Celje d.o.o., Celje, Slovenia. The USP device uses a custom-made ultrasonic generator with a 1.6 MHz ultrasonic transducer LIQUIFOG II (Johnson Matthey Piezo Products GmbH, Redwitz, Germany) as its base [29,30,31]. The ultrasonic generator produces aerosol droplets from the Au precursor, which are transported by carrier gas (N_2_) into a heating furnace, which is composed of 3 zones. The first zone is evaporative, and the second zone is reactive, where H_2_ enters, resulting in thermal decomposition of the dried Au-ions and in reduction in the Au^3+^ into Au^0^ as AuNPs. The sintering of AuNPs is carried out in the third zone. AuNPs are finally collected in four serially connected gas washing bottles in D.I water and PVP (1 g/L)—as presented in Figure 1.

For preparation of both precursors, two key Au concentrations (0.5 and 1 g/L) were chosen, that represented four different types of precursors: AuAc 0.5 g/L, AuAc 1 g/L, AuCl 0.5 g/L, and AuCl 1 g/L (these names are used throughout the text). The temperature (T_1_, T_2_, and T_3_) and gas (N_2_ and H_2_) flows in the heating furnace were consistent with previous experiments [27,28]. The detailed parameters of the USP synthesis are the subject of an applied patent for AuNP production [32]. The volume of each collected AuNPs’ suspension was 1 L.

#### 2.2.2. Preparation of Gold Inks

Each AuNPs’ suspension was concentrated using Rotavopor distillation (Rotavopor (BUCHI Rotavapor R-300, BÜCHI Labortechnik AG, Flawil, Switzerland) with the same parameters. The distillation parameters were: (1) pressure—40 mBar, (2) speed of rotation of the evaporating flask—240 rpm, (3) heating bath temperature—40 °C, (4) the initial volume of suspension—250 mL × 4, (5) 1000 mL of each suspension was concentrated to 50 mL and the final volume of suspension—12.5 mL × 4, and (6) distillation time—1 h × 4 = 4 h. Figure 2 shows the prepared gold inks AuAc 1 g/L and AuCl 1 g/L. Colloidal stability of the inks was evaluated by turbidity measurements (TB1 Turbidimeter, Velp Scientifica, Usmate Velate (MB), Italy) over a period of 4 weeks in which the inks remained stable, hence appropriate for ink-jet printing. The AuCl 1 g/L, AuAc 1 g/L, AuCl 0.5 g/L, and AuAc 0.5 g/L inks exhibited “ejectability” numbers of 5.75, 5.80, 5.82, and 5.65 1/Oh, respectively. 

#### 2.2.3. Inkjet Printing

A Dimatix Fujifilm inkjet Printer DMP-2831 (Fujifilm, Santa Clara, CA, USA) with a piezoelectric ink cartridge (with a drop volume of 10 pL) equipped with a fiducial camera, was used for the testing of inject printing of gold inks onto the photo paper. Characteristic parameters were: jetting voltage for all nozzles: 23 V, tickle control (frequency): 23 kHz, head angle: 4.5°, cartridge temperature: 28 °C; platen temperature: 60 °C. The choice of photo paper was random as it was available in the laboratory. The printed pattern dimensions were 60 mm × 9 mm (Figure 3). 

### 2.3. Characterisation

#### 2.3.1. Inductively Coupled Plasma-Mass Spectrometry (ICP-MS)

The concentration of synthesised AuNPs in all four prepared gold inks was measured with inductively coupled plasma-mass spectrometry (ICP-MS). The spectrometer used was an HP, Agilent 7500 CE, equipped with a collision cell (Santa Clara, CA, USA). The following conditions were used for the ICP-MS: the power was 1.5 kW, Nebuliser-Meinhard, plasma gas flow was 15 L/min, nebuliser gas flow was 0.85 L/min, make up gas flow was 0.28 L/min, and reaction gas flow was 4.0 mL/min. The instrument was calibrated with matrix-matched calibration solutions. The relative measurement uncertainty was estimated as ±3%.

#### 2.3.2. Dynamic and Electrophoretic Light Scattering (DLS)

A Zetasizer Nano ZS device (Malvern Instruments Ltd., Malvern, UK) was used to determine the mean hydrodynamic diameter (d nm) and the ζ-potential of the AuNPs in gold inks by dynamic (DLS) and electrophoretic light scattering (ELS) of the particles. The DLS measurements were conducted using a 633 nm light source and a scattering angle of 173°. Slow and fast field reversal measurements were performed in the electrophoretic mobility experiments. All measurements were performed at 25 °C using water (RI = 1.33, viscosity = 0.8872 cP, and dielectric constant = 78.5) as the dispersant for calculation of the results. Measurements were performed in triplicates.

#### 2.3.3. UV–VIS Spectrophotometry

A Tecan Infinite M200 UV/VIS Spectrophotometer (Tecan, Salzburg, Austria), using a quartz cuvette, was used for the UV/VIS absorption, measured over the wavelength range of 400–750 nm, with number of flashes = 5× and time per measure = 20 ms.

#### 2.3.4. Scanning Electron Microscopy

A scanning electron microscope, Sirion 400NC (FEI, **FEI**, Hillsboro, OR, USA), with an energy-dispersive X-ray spectroscope, INCA 350 (Oxford Instruments, Oxford, UK), was used for the SEM/EDS analysis. It had a Schottky electron source, where the field emission produces a jet of electrons with a small diameter and a high density. The result is high resolution, even at low voltages: 1.0 nm at 15 kV and 2.0 nm at 1 kV. The AuNPs’ suspensions were put dropwise onto the SEM holders (mesh) with conductive carbon adhesive tape, which allowed better SEM observation. The SEM holders were left to dry in a desiccator for 1 day before the SEM investigations were carried out.

#### 2.3.5. Optical Goniometry

An optical goniometer, OCA 35 (DataPhysics, Filderstadt, Germany), was used to measure the time-dependent static contact angle of the water (SWCA) and static contact angles (SCA) of all four gold inks with varying surface tensions (water, ethylene glycol, formamide, and diiodomethane) on the surfaces of gold inks’ printed photo paper. The time-dependent measurements (t) were conducted automatically at a time rate of 1 s^−1^, and the contact angle (θ) was evaluated by ellipse fitting. A drop volume of 3 µL was used to measure the contact angles at three different spots on the photo paper. Measurements were conducted under ambient conditions (T = 25 °C, relative humidity = 55%). The surface free energy ($\gamma_{s}^{TOT}$) of the photo papers was calculated using the SCA values of the four chosen liquids by the acid–base approach of Good, van Oss, and Chaudhury (GvOC), which divides the $\gamma_{s}^{TOT}$ into the dispersive Lifshitz–van der Waals interactions ($\gamma_{s}^{LW}$) and the polar Lewis acid–base interactions ($\gamma_{s}^{AB}$), as shown in Equation (1). A detailed description of the theory can be found elsewhere [33].
(1)$$\gamma_{s}^{TOT}=\gamma_{s}^{LW}+\gamma_{s}^{AB}=\gamma_{s}^{LW}+2\left(\sqrt{\gamma_{s}^{+}\gamma_{s}^{-}}\right),$$
where $\gamma_{s}^{+}$ and $\gamma_{s}^{-}$ are the electron acceptor (Lewis acid) and electron donor (Lewis base) components, respectively. Combined with the Young’s equation, the relationship for the work of adhesion between a liquid and a solid can be derived as:(2)$$W_{adh}=\gamma_{L}\left(1+cos\theta \right)=2\left(\sqrt{\gamma_{L}^{LW}\gamma_{s}^{LW}}\right)+2\left(\sqrt{\gamma_{L}^{+}\gamma_{s}^{-}}\right)+2\left(\sqrt{\gamma_{L}^{-}\gamma_{s}^{+}}\right),$$
where the subscripts *L* and *s* refer to liquid and solid, respectively. The time-dependent SCA (t) of individual inks on the neat photo paper were measured additionally, to observe their wetting of the photo paper in comparison to pure water.

#### 2.3.6. Statistics

ImageJ software (version 1.51, NIH, Bethesda, MD, USA) was used for the size measurements of AuNPs from the SEM micrographs. A total of 1000 AuNPs were measured for each of the gold inks. Bin sizes of 10 and 5 nm were used for the size distribution representations. Mean values were calculated from the measured particle sizes for each sample for the SEM investigation. The mean values and standard deviations for the DLS size measurements with distributions and ζ-potential measurements of AuNPs were given by the Malvern Zetasizer measurement software 7.12, (Malvern Instruments Ltd., Malvern, UK)

## 3. Results and Discussion

### 3.1. ICP-MS

The measured final concentrations of Au in all the prepared gold inks are shown in Table 1. We found out that higher concentrations of Au in the precursor allowed the formation of gold ink with a higher final AuNPs’ content under the same conditions. It was discovered that, under the same gold ink preparation conditions, the final Au concentration was higher in both AuAc precursors in comparison with AuCl. This can be attributed to the lower losses of AuNPs in the USP synthesis process from AuAc precursors and the formation of more favourable and stable AuNPs’ structures in suspensions. On the other hand, there was no detectable effect of ionic strength on the final concentration of AuNPs in gold inks, which is consistent with previous research [34]. 

### 3.2. AuNPs’ Size and ζ-Potential of Gold Inks

As can be seen in Figure 4, the AuNPs in the prepared gold inks exhibit Z-average hydrodynamic diameters ranging from 99.09 ± 3.24 nm (AuCl 1 g/L) to 115.92 ± 1.87 nm (AuCl 0.5 g/L). The rather insignificant differences in Z-average values between different samples show that neither the type of precursors, nor the AuNPs’ concentration influences their average hydrodynamic diameter in the gold inks. The Z-average range for all measured inks was suitable for inkjet printing, as the diameter of the printing nozzles was 21 µm [35]. However, the polydispersity index (PDI) of the AuNPs in the gold inks was rather high, as it ranged from 0.279 (AuCl 0.5 g/L) to 0.403 (AuAc 0.5 g/L). The PDI values seemed consistent with data reported by other authors [36]. Nevertheless, the size, shape, and PDI can vary greatly, depending on the PVP concentration during nanoparticle preparation [37]. Particles of over 1 µm in size were detected as well, which poses a problem for ink jet printing, as agglomerates of µm sizes can cause nozzle blockage [35]. Therefore, prior to ink jet printing, all inks were filtered through a PTFE syringe filter with a pore diameter of 0.1 µm. Interestingly, when looking at the ζ-potential data, one can observe low average values ranging from −1.51 ± 25.34 mV (AuCl 1 g/L) to −2.89 ± 20.68 mV (AuCl 0.5 g/L). The influence of increasing PVP concentration and molecular weight is known to reduce the absolute ζ-potential values of negatively charged inorganic particles [38]. This agrees with the concentration and molecular weight of the PVP used in this work. Similarly, low ζ-potential values were obtained by other authors as well [36]. The high standard deviation of the ζ-potential values reflects a broad ζ-potential distribution, resulting from the broad distribution of electrophoretic mobility, hence, charges in the inks. This can be a contribution of the negatively charged free PVP, PVP neutralised AuNPs, and the cationic nature of uncoated dissociated AuNPs [36].

### 3.3. UV–VIS

The various states of the PVP and AuNPs in gold inks were not studied further in this work. Figure 5 shows the UV–VIS spectra of the prepared gold inks within the range of 510–560 nm [39], which exhibits their unique optical properties, along with a property known as surface plasmon resonance.

### 3.4. SEM Analysis of AuNPs

Figure 6 shows the SEM micrographs of the AuNPs taken from suspensions synthesised through USP. The image inserts show the AuNPs’ morphologies, and their size distribution is given for comparison of all suspensions. The size distributions were composed of 1000 ImageJ AuNPs’ size measurements for each sample, obtaining a general overview of the particle sizes. The acquired size distributions are in agreement with the sizes given from DLS measurements, with small deviations between the results of the two measurement methods. From the SEM micrographs, there is also a well visible difference in the AuNPs’ morphology. AuNPs in AuCl suspensions have mostly spherical and irregularly shaped particles; AuAc 1 g/L has similar AuNPs shapes, mostly spherical with some irregular particles. AuAc 0.5 g/L has more irregular AuNPs present, along with spherical particles, while the irregular particles are also somewhat bigger than the spherical ones. The measured size distribution for this sample was even more erratic, not showing a clear peak for the sizes of the AuNPs that were measured. The irregular AuNPs were also more difficult to measure in a relevant way, as they had different dimensions in a single shape. The largest dimensions of these AuNPs were chosen for the measurements. 

After filtration of the AuNPs in suspensions through a PTFE syringe filter, the SEM investigation was repeated for the samples, as shown in Figure 7. A much higher number of smaller AuNPs was detected in the AuCl samples’ measurements than with the AuAc samples. The filter pore diameter of 0.1 µm dictated that the filtered AuNPs should have a size below 100 nm; however, it seems to have also been more effective for the AuCl samples, acting as a barrier for all but the smallest AuNPs’ sizes. This resulted in smaller mean sizes of about 20 nm, and a sharper, more narrow size distribution as compared to the AuAc samples. This outcome of filtration may be due to the different chemical constituents remaining in the suspensions produced with USP and the surface charges of the resulting particles. An important note is that only 1000 AuNPs were measured from the samples, and this investigation showed an overview of the sample conditions. However, it was evident from this investigation that although the larger AuNPs may also have been present in the AuCl samples, they were present in far fewer numbers compared to particles of 50 nm and lower.

### 3.5. Contact Angle and Surface Energy Measurements of Gold Inks

As can be observed in Figure 8A, the SWCA(t) of the inkjet-printed AuNPs on photo paper showed a faster rate of change of SWCA over time than the neat photo paper. The SWCA(t) of neat photo paper changed from the initial 56.0° (at 0 s) to 51.0° (at 70 s) at an average rate of 0.07 ± 0.09°/s. This can be attributed largely to water evaporation from the drop, and less to absorption and wetting of the paper. This was confirmed by running the same experiment on a non-absorptive surface, i.e., a silicon wafer (Figure 8A). The contact angle, in this case, changed from the initial 65.0° (at 0 s) to 59.8° (at 70 s) at an average rate of 0.07 ± 0.05. This is a reflection of evaporation, as water cannot get absorbed by the non-absorptive wafer. The same can be concluded for water on photo paper, as the rate of SWCA (t) change was the same as in the case of the silicon wafer. On the contrary, the SWCA(t) of AuCl 1 g/L changed at a rate of 2.23 ± 1.73°/s in the first 10 s from 37.4° to 18.5°, indicating wetting of the printed paper by water molecules over time. It is worth noticing that, in the first 3 s, the rate changed at the highest rate of 5.13°/s, which reflects the fast-spreading of water on the AuNPs’ print. After that, the SWCA(t) changed at an average rate of 0.13 ± 0.13°/s, indicating that water evaporation was driving the SWCA(t) change instead of paper wetting. A similar trend was observed in the case of AuAc 1 g/L, where the SWCA(t) changed at an average rate of 1.23 ± 0.50 °/s in the first 10 s from 35.3° to 24.2°. The highest rate in the first 3 s was 1.99°/s, which was more than 2-fold lower than in the case of AuCl gold inks. Although the rate was lower when compared to AuCl 1 g/L, wetting of the gold inks-printed paper can be confirmed. The rate dropped to an average of 0.26 ± 0.22°/s between 10 and 70 s. Similar behaviour can be observed at lower ink concentrations, although the initial SWCA(t) rates were lower, which points to a slower wetting of the printed paper than in the case of 1 g/L ink. This can be related to thinner prints in the case of 0.5 g/L (h = mm) than in 1 g/L (h = mm). In the case of the AuCl 0.5 g/L ink, the SWCA(t) changed at an average rate of 0.71 ± 0.42°/s in the first 10 s from the initial 23.8° to 11.4°, followed by an average rate of 0.10 ± 0.09°/s. In the case of the AuAc 0.5 g/L ink, the SWCA(t) changed at an average rate of 0.56 ± 0.20°/s in the first 10 s from initial 26.9° to 15.3°, followed by a rate of 0.11 ± 0.08°/s between 10 and 70 s. Drop images of the optical SWCA(t) measurements at 0 and 70 s for the neat and gold-printed photo paper are shown in Figure 8B. The results suggest that water wetting of the gold inks-printed patterns can be controlled by varying the concentration of AuNPs in the gold ink.

The wetting of the photo paper by the gold ink itself was also studied, and the SWCA(t) and the drop images are shown in Figure 8C. The SCA(t) of the water drop changed at an average rate of 0.07 ± 0.09°/s, and it changed at a similar rate of 0.06 ± 0.05°/s, 0.07 ± 0.05°/s, 0.07 ± 0.05°/s, and 0.07 ± 0.05°/s for the AuCl 0.5 g/L, AuCl 1 g/L, AuAc 0.5 g/L, and AuAc 1 g/L, respectively. Thus, the AuNPs in the aqueous inks did not change the water wetting behaviour of photo paper, even though the initial contact angles (0 s) were slightly lower than in the case of water. The lower initial contact angles can be attributed to the higher dispersive part of the AuNPs ($\gamma_{s}^{LW}$ = 35.4–41.3 mN/m) in comparison with water ($\gamma_{s}^{LW}$ = 21.8 mN/m), a contribution of the stabilising polymer PVP (Table 2 PVP). The higher dispersive part value was closer to the value of neat photo paper (42.6 mN/m), making the wettability of the inks slightly higher than water. The surface free energy of PVP was determined by Lee to be 48.5 mN/M and by Van Oss to be 43.4 mN/M (Table 2). It exhibited high dispersive ($\gamma_{s}^{LW}$) interactions (43.4 mN/m) and rather a high electron-donating Lewis-base part ($\gamma_{s}^{-}$ = 15.3–29.7 mNm). When looking at the photo paper itself, the SFE of 42.6 mN/m was very similar to PVP. A slightly lower dispersive part was observed ($\gamma_{s}^{LW}$ = 33.8 mN/m). The neat gold exhibited an SFE of 36.5 mN/m, which was lower than the PVP and the photo paper. It also exhibited a dispersive part ($\gamma_{s}^{LW}$ = 33.5 mN/m) and the Lewis-base part ($\gamma_{s}^{-}$ = 19.3 mNm) values similar to the photo paper.

Furthermore, when looking at Figure 8D, we can see that the gold ink jet-printed patterns exhibited $\gamma_{s}$ values of 41.3 ± 0.9, 41.7 ± 1.9, 38.4 ± 1.0, and 35.4 ± 2.2 for AuAc 1 g/L, AuAc 0.5 g/L, AuCl 1 g/L, and AuCl 0.5 g/L, respectively. These values were 5–27% lower than the theoretical values for PVP. This indicated that the interactions of the printed patterns and the liquid occurred partially on the PVP–liquid interface (higher $\gamma_{s}$) and partially on the gold–liquid interface (lower $\gamma_{s}$). This confirmed the zeta potential measurements, assuming that the inks contained a variety of charges. The increasing Lewis-base component was difficult to evaluate, as the GvOC approach overestimated it, which arose mostly from the reference values for the dispersive and polar parts used for water [32]. Therefore, the absolute values of the base parts are not discussed in greater detail.

### 3.6. SEM Analysis of Printing Pattern

The printing pattern characterisations are presented in Figure 9. SEM micrographs confirmed the presence of AuNPs on the surface of the photo paper. The formed prints showed a good distribution of AuNPs over the surface of the photo papers. Evidently, a significantly higher concentration of AuNPs was observed in those where gold inks formed from precursors with an initial higher concentration of Au = 1 g/L were used, which was also consistent with the obtained results of ICP-MS and other analyses.

SEM/EDX analyses identified the presence of Au, which is a direct confirmation of the AuNPs, while identification of C, O, Al, and Si was sourced from the photo paper with a glossy coating, which was used as the testing printing substrate for the gold inks.

## 4. Conclusions

The following conclusions can be drawn from the current research work:The USP synthesis, in combination with the rotavapor, allowed successful preparation of gold inks with AuNPs reaching Au concentrations more than 300 ppm.SEM observation revealed that AuNPs in the as prepared inks had mostly spherical shape, with the presence of some irregularly shaped AuNPs.AuNPs in the prepared inks exhibited Z-average hydrodynamic diameters, ranging from 99 to 115 nm.Due to the fact that some AuNPs were over 1 µm in size, consequently, the filtration of as prepared gold inks was mandatory to avoid problems with the inkjet printing, as AuNPs agglomerated, which can cause nozzle blockage.After filtration, the AuNPs in gold inks had an average size between 20 and 46 nm, suitable for inkjet printing. The AuNPs size was directly dependent on the Au concentration in the precursor (lower AuNPs’ size–lower Au concentration).The measured UV–VIS spectra of as prepared gold inks were within the range of 510–560 nm.The contact angle and surface energy results showed that the printed patterns could be controlled by varying the concentration of AuNPs in the inks.The filtered gold inks were printed on photo paper with selected parameters, and it confirmed relative homogeneous distribution of AuNPs on their surfaces. Additional EDS measurements showed the presence of Au.

## Figures and Tables

**Figure 1 nanomaterials-11-00599-f001:**
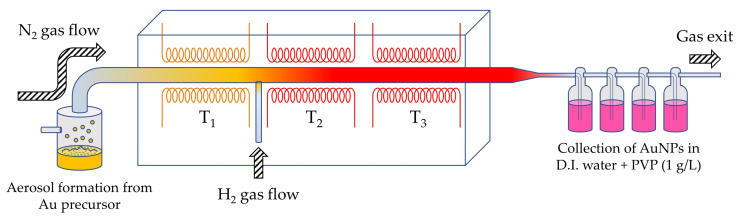
A schematic presentation of the ultrasonic spray pyrolysis (USP) device used for the synthesis of gold nanoparticles (AuNPs).

**Figure 2 nanomaterials-11-00599-f002:**
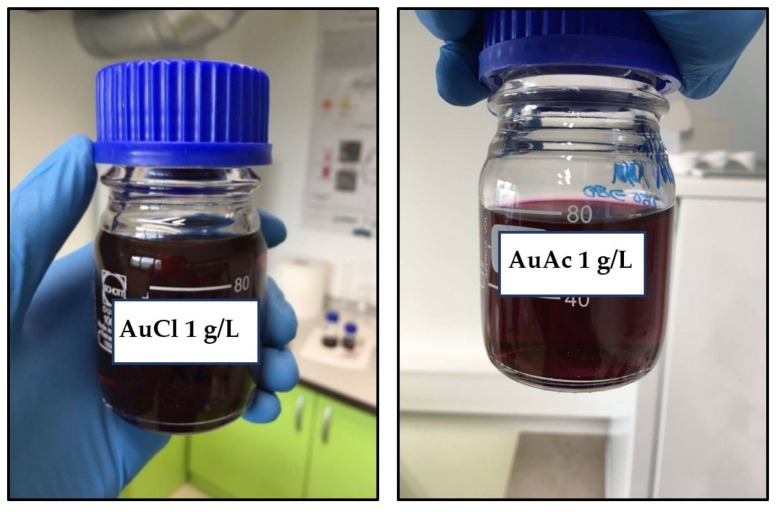
Prepared gold inks gold (III) chloride tetrahydrate (AuCl) and Au acetate salt (AuAc) with Au concentration of 1 g/L in the precursor.

**Figure 3 nanomaterials-11-00599-f003:**
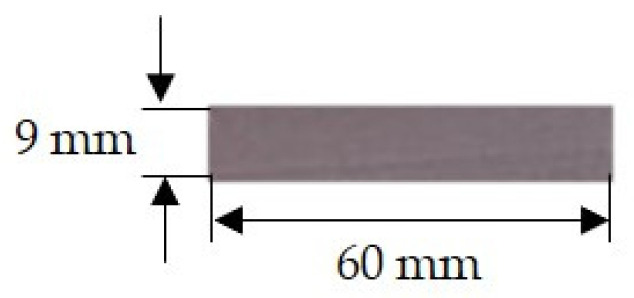
Presentation of gold ink-printed pattern on photopaper.

**Figure 4 nanomaterials-11-00599-f004:**
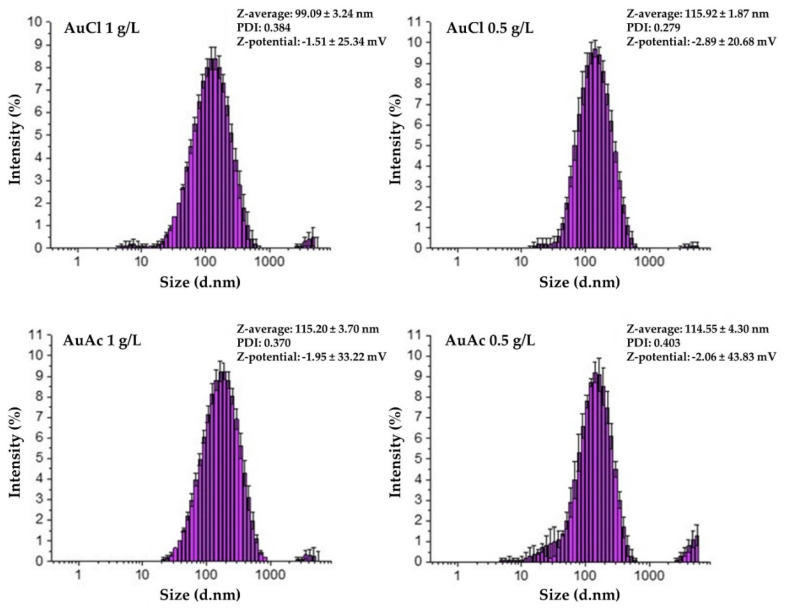
AuNPs’ size (hydrodynamic diameter) distribution by intensity, polydispersity index (PDI), and ζ-potential values for the as prepared gold inks.

**Figure 5 nanomaterials-11-00599-f005:**
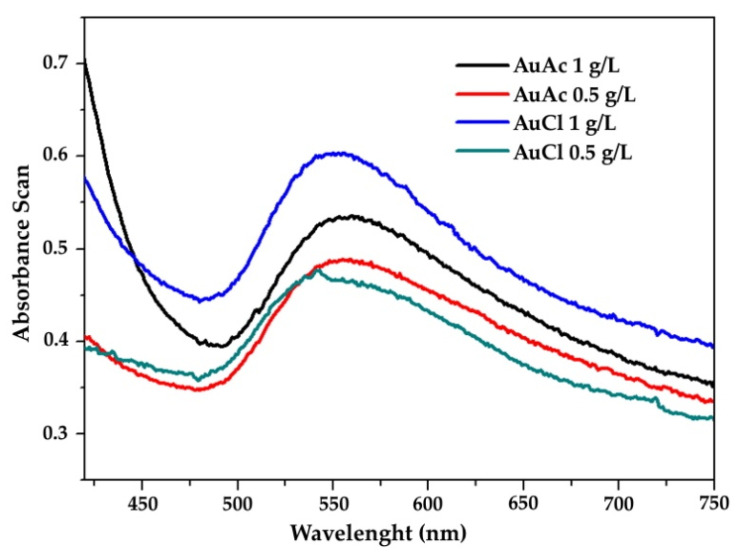
UV–Vis spectra of different concentrations of AuNPs.

**Figure 6 nanomaterials-11-00599-f006:**
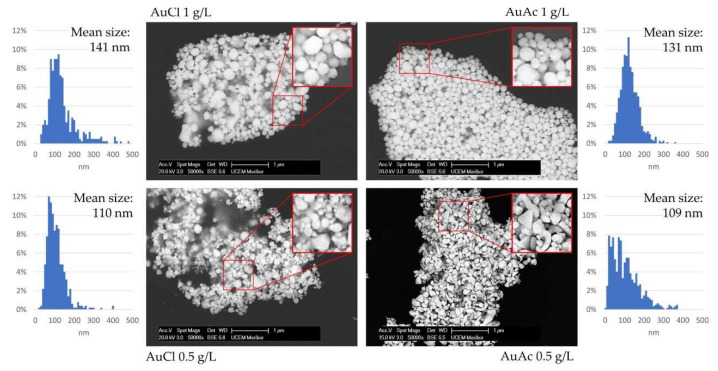
SEM micrographs of AuNPs taken from suspensions synthesised through USP, with corresponding size distributions from ImageJ.

**Figure 7 nanomaterials-11-00599-f007:**
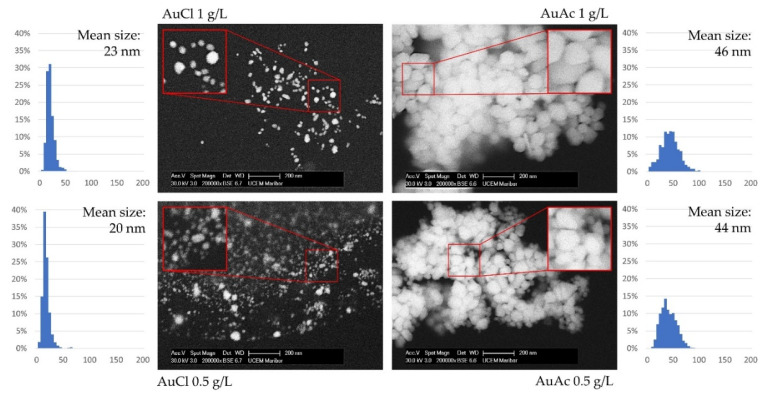
SEM micrographs of AuNPs taken from filtered suspensions, with corresponding size distributions from ImageJ.

**Figure 8 nanomaterials-11-00599-f008:**
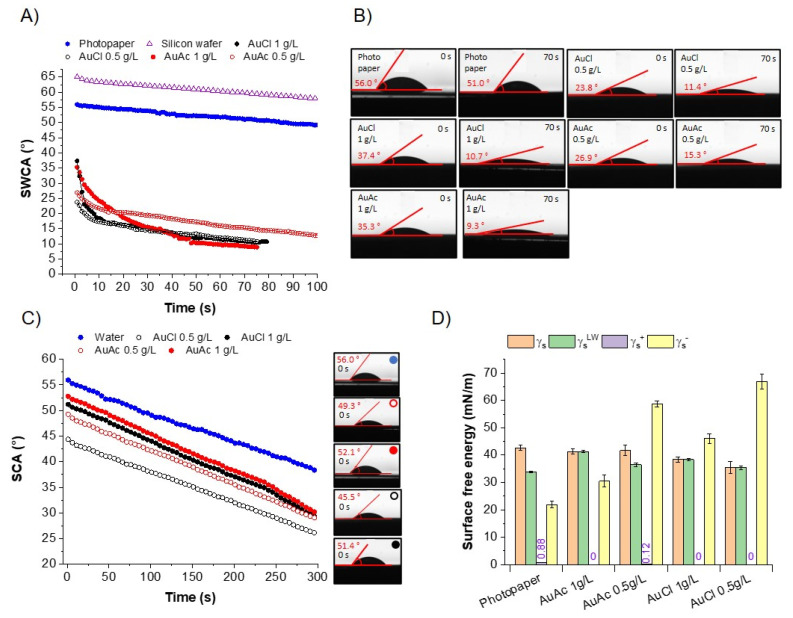
OCA: Optical contact angle measurements of inkjet-printed patterns from gold inks. (**A**) Static contact angle of the water (SWCA)(t) on neat photo paper and inkjet-printed patterns of gold inks (lines represent polynomial fits; order = 9); (**B**) drop images of water from SWCA(t) measurements at the time of contact (0 s) and after 70 s; (**C**) static contact angles (SCA)(t) of water and gold inks on photo paper; and (**D**) surface free energy calculations of the inkjet-printed patterns from gold inks using the Good, van Oss, and Chaudhury (GvOC) approach.

**Figure 9 nanomaterials-11-00599-f009:**
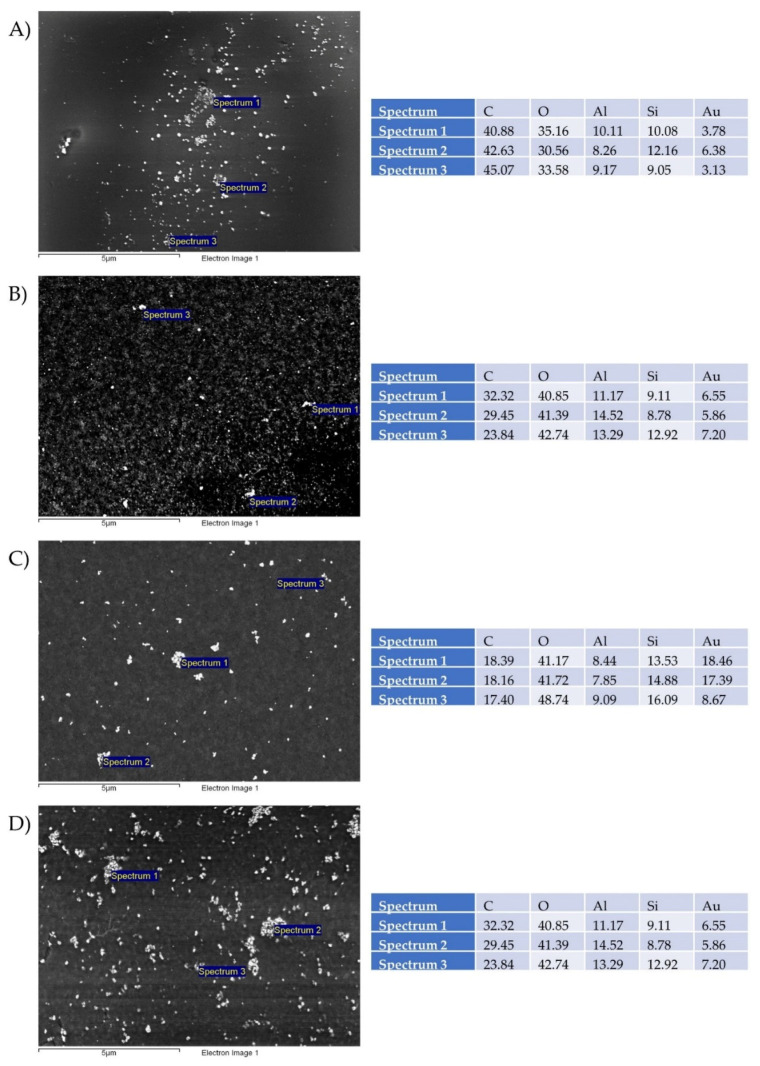
SEM/EDS studies of printed patterns on photo paper with precursor concentrations of: (**A**) AuCl 1 g/L, (**B**) AuCl 0.5 g/L, (**C**) AuAc 0.5 g/L, and (**D**) AuAc 1 g/L. The values are presented in weight percentages.

**Table 1 nanomaterials-11-00599-t001:** Au concentration in the prepared gold inks.

Gold Ink	Au Concentration (ppm)
AuCl 0.5 g/L	300
AuCl 1.0 g/L	400
AuAc 0.5 g/L	340
AuAc 1.0 g/L	450

**Table 2 nanomaterials-11-00599-t002:** SFE polyvinylpyrrolidone (PVP): surface free energy of photo paper (determined experimentally), dental gold type III, and polyvinylpyrrolidone as reported in the literature.

Author	Model	$\gamma_{s}$ (mN/m)	$\gamma_{s}^{LW}$	$\gamma_{s}^{AB}$	$\gamma_{s}^{+}$	$\gamma_{s}^{-}$
PVP by Ma et al. [13]	GvOC	48.5	43.4	5.1	0.4	15.3
PVP by Van Oss et al. [40]	GvOC	43.4	43.4	0.0	0.0	29.7
Gold by Knorr et al. [41]	Least square analysis	36.5	33.5	3.01	0.17	19.3
Photo paper (this study)	GvOC	42.6	33.8		0.9	21.9

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
