# Peer review of "Gold Inks for Inkjet Printing on Photo Paper: Complementary Characterisation"

_nanomaterials, 2021, doi:10.3390/nano11030599_

Round 1

Reviewer 1 Report

This paper describes a study on the preparation of inkjet inks based on gold nanoparticles.

The title of the paper lacks of precision, as the paper mainly deals with the surface energy of the inks, and not at all on the other properties required for an inkjet ink, for instance viscosity (in mPa.s, preferably to cP), volumic mass, and more generally ejectability criteria.

 The introduction presents the context of the study and gives a short state-of-the-art on gold inks used in printed electronics.

The materials and methods of the research are generally correctly described, except for the inkjet printing part (§2.2.3). For instance, the parameter “Waveform: low viscosity” is not explicit. In addition, the dimensions of the printed patterns are not mentioned. Consequently, it is difficult for the reader to figure out what is represented on Figure 8.

The discussion on the surface energy is well-conducted.

As mentioned above, the reader would appreciate more information about the other properties of the ink. For instance, the calculation of the “ejectability” number (1/Oh) would be relevant in a paper devoted on inkjet inks. In addition, one could expect that typical inkjet-printing problem such as “coffee ring” effect should have been evoked.

Please check the errors of English.

Author Response

A detailed list of explicit corrections and additions is attached below. All corresponding changes in the corrected version of the manuscript are highlighted in red.

Reviewer 1

the title of the paper lacks of precision, as the paper mainly deals with the surface energy of the inks, and not at all on the other properties required for an inkjet ink, for instance viscosity (in mPa.s, preferably to cP), volumic mass, and more generally ejectability criteria.

New title: Gold Inks for Inkjet printing on photo paper: Complementary Characterisation”.

The introduction presents the context of the study and gives a short state-of-the-art on gold inks used in printed electronics.

The materials and methods of the research are generally correctly described, except for the inkjet printing part (§2.2.3). For instance, the parameter “Waveform: low viscosity” is not explicit. In addition, the dimensions of the printed patterns are not mentioned. Consequently, it is difficult for the reader to figure out what is represented on Figure 8.

A Figure 3 has been inserted in the manuscript aditionally showing a photograph of the printed pattern and its dimensions for clarification. The “waveform: low viscosity” has been removed from the text, this was named as our internal reference, which was the customised waveform used in the study.

Figure 8 – now Figure 9 shows the presence and distribution of gold inks printed on the photo paper.

The discussion on surface energy is well-conducted.

As mentioned above, the reader would appreciate more information about the other properties of the ink. For instance, the calculation of the “ejectability” number (1/Oh) would be relevant in a paper devoted on inkjet inks. In addition, one could expect that typical inkjet-printing problem such as “coffee ring” effect should have been evoked.

Ejectability numbers were calculated for all prepared inks. The values range between 5.5. and 6, and have been inserted in the manuscript in lines 93-95. The “coffee ring” was not observed in the printed patterns (scheme 1 added in the manuscript), therefore, further elaboration of the “coffee ring” effect was not performed.

Please check the errors of English – The manuscript has been checked again by Shelagh Baker, a native speaker.

Reviewer 2 Report

The manuscript by Tiyyagura et al. described synthesis of gold nanoparticles and gold inks for inkjet printing application on photo paper. Although some data look interesting, it needs corrections before consideration of its publication in Nanomaterials. The followings are some comments to revise the manuscript.

(1) The authors already published some papers regarding gold nanoparticle synthesis using the same technique. Therefore, the authors should emphasize the novelty and difference of this manuscript compared with the previous papers.

(2) In “2.2.1. Preparation of AuNPs”, the authors did not mention the used temperatures (T1, T2, and T3) and gas flow rates (N2, and H2) with some references. Detailed experimental conditions should be written.

(3) From Figure 1, it looks like that the gas flows through the upper parts of the collection bottles, without passing through the collecting liquid. Fig. 1 should be redrawn.

(4) After filtering the suspensions, the AuNPs showed quite different sizes. Therefore, the yields of AuNPs before and after filtering should be compared.

(5) The stability of AuNPs should be tested for inkjet printing applications.

Author Response

A detailed list of explicit corrections and additions is attached below. All corresponding changes in the corrected version of the manuscript are highlighted in red.

Reviewer 2

The manuscript by Tiyyagura et al. described synthesis of gold nanoparticles and gold inks for inkjet printing application on photo paper. Although some data look interesting, it needs corrections before consideration of its publication in Nanomaterials. The followings are some comments to revise the manuscript.

(1) The authors already published some papers regarding gold nanoparticle synthesis using the same technique. Therefore, the authors should emphasize the novelty and difference of this manuscript compared with the previous papers.

The USP method is capable of producing gold inks from various precursor salts, resulting in inks with differing AuNPs` morphology. In previous studies there were gaps in the literature and research, namely, the gold inks' properties and their usability in inkjet printing. The present study highlights these properties for determining the most optimal gold ink for printing, produced by USP.

(2) In “2.2.1. Preparation of AuNPs”, the authors did not mention the used temperatures (T1, T2, and T3) and gas flow rates (N2, and H2) with some references. Detailed experimental conditions should be written.

All of the detailed parameters of the USP synthesis were not disclosed in the text, as they are the  subject of an applied patent for AuNP production (added reference 32). This has now been clarified in the text under Chapter 2.2.1. Preparation of AuNPs.

(3) From Figure 1, it looks like that the gas flows through the upper parts of the collection bottles, without passing through the collecting liquid. Fig. 1 should be redrawn

The Figure 1 has been corrected to show the gas flow passing through the collection liquid.

(4) After filtering the suspensions, the AuNPs showed quite different sizes. Therefore, the yields of AuNPs before and after filtering should be compared

The concentration of AuNPs was measured before (data are not shown in the manuscript) and after filtering. The loss of AuNPs after filtering was approximately 60 % in each case. The AuNPs lost due to filtering can be recovered and reused for other applications e.g. decorative gold coatings.

(5) The stability of AuNPs should be tested for inkjet printing applications

The colloidal  stability of the AUNPs in the ink was tested by means of turbidity measurements over a period of 1 week. The inks remained stable during this time. We have included this sentence in the manuscript (lines 95-97).

Reviewer 3 Report

In this paper the authors developed gold nanoparticles -based gold ink by UPS method for inject printing on photo paper. This work may have potential in electronic industry due to formation of cost-effective flexible paper-based electronics. However, I do not recommend for publications in “Nanomaterials” due to lack of novelty. There similar papers were published in different journals, for example, references 27 and 28 of this manuscript.

Additional comments:

  • The authors may rethink about the title of the paper. This title seems to be title of a review paper.
  • Minor spell check is required.
  • They can include cyclic voltammetry of the printed paper. This will be useful for the researchers who work on electrochemical sensors.

Author Response

A detailed list of explicit corrections and additions is attached below. All corresponding changes in the corrected version of the manuscript are highlighted in red.

Reviewer 3

In this paper the authors developed gold nanoparticles -based gold ink by UPS method for inject printing on photo paper. This work may have potential in electronic industry due to formation of cost-effective flexible paper-based electronics. However, I do not recommend for publications in “Nanomaterials” due to lack of novelty. There similar papers were published in different journals, for example, references 27 and 28 of this manuscript.

The present work aims to prepare gold inks and examine their surface characterisation, different concentrations and their printability, which are not available in the current literature. We strongly believe that this work is valuable information for cost-effective, flexible paper-based electronics.

Additional comments:

  • The authors may rethink about the title of the paper. This title seems to be title of a review paper.

Thank you for the comment, and we agree with the reviewer. We have changed the title of the paper according to the suggestion: New title: Gold Inks for Inkjet printing on photo paper: Complementary Characterisation”.

  • Minor spell check is required – English checking has been done again by Shelagh Hedges, a native speaker (We have checked and polished all the typos and the language in the revised manuscript carefully).
  • They can include cyclic voltammetry of the printed paper. This will be useful for the researchers who work on electrochemical sensors.

We appreciate the reviewer`s suggestions regarding the cyclic voltammetry test—however, this analysis is beyond the scope of the current objective. So we will perform these electrochemical tests, like impedance and cyclic voltammetry tests, in our future studies of electrochemical sensors.

Round 2

Reviewer 1 Report

Thank you for having taken into account my comments.

Reviewer 2 Report

The manuscript was properly revised based on the reviewer's comment. I am in favor of publishing this manuscript in Nanomaterials.

Reviewer 3 Report

The authors have addressed my concerns and improved the paper. This paper can be considered for publication in its present form.